# Management Recommendation Generation for Areas Under Forest Restoration Process through Images Obtained by UAV and LiDAR

**Bruna Paolinelli Reis** [1,2,*], **Sebastião Venâncio Martins** [1], **Elpídio Inácio Fernandes Filho** [3], **Tathiane Santi Sarcinelli** [4], **José Marinaldo Gleriani** [1], **Gustavo Eduardo Marcatti** [5], **Helio Garcia Leite** [1] and **Melinda Halassy** [6]

1. Forest Engineer Department, Universidade Federal de Viçosa, Avenida P. H. Rolfs s/n, Viçosa 36570-000, MG, Brazil
2. Eötvös Loránd University, Department of Plant Taxonomy, Ecology and Theoretical Biology, Pázmány P. stny. 1/C, 1117 Budapest, Hungary
3. Soil Department, Universidade Federal de Viçosa, Avenida P. H. Rolfs s/n, Viçosa 36570-000, MG, Brazil
4. Suzano S.A., Meio Ambiente Florestal, Rodovia Aracruz—Barra do Riacho, s/n, km 25, Aracruz 29197-900, ES, Brazil
5. Department of Agricultural Sciences, Universidade Federal de São João Del Rei, Rodovia MG 424—km 47„ Sete Lagoas 35701-9970, MG, Brazil
6. MTA Centre for Ecological Research, Institute of Ecology and Botany, Alkomány u. 2-4, 2163 Vácrátót, Hungary
* Correspondence: bruna.paolinelli@okologia.mta.hu

**Abstract:** Evaluating and monitoring forest areas during a restoration process is indispensable to estimate the success or failure of management intervention and to correct the restoration trajectory through adaptive management. However, the field measurement of several indicators in large areas can be expensive and laborious, and establishing reference values for indicators is difficult. The use of supervised classification techniques of high resolution images, combined with an expert system to generate management recommendations, can be considered promising tools for monitoring and evaluating restoration areas. The objective of the present study was to elaborate an expert system of management recommendation generation for areas under restoration, which were monitored by two different remote sensors: UAV (Unmanned Aerial Vehicle) and LiDAR (Light Detection and Ranging). The study was carried out in areas under restoration with about 54 ha and five years of implementation, owned by Fibria Celulose S.A. (recently acquired by Suzano S.A.), in the southern region of Bahia State, Brazil. We used images from Canon S110 NIR (green, red, near infrared) on UAV and LiDAR data compositions (intensity image, digital surface model, digital terrain model, normalized digital surface model). The monitored restoration indicator entailed land cover separated into three classes: Canopy cover, bare soil and grass cover. The images were classified using the Random Forest (RF) and Maximum Likelihood (ML) algorithms and the area occupied by each land cover classes was calculated. An expert system was developed in ArcGIS to define management recommendations according to the land cover classes, and then we compared the recommendations generated by both algorithms and images. There was a slight difference between the recommendations generated by the different combinations of images and classifiers. The most frequent management recommendation was "weed control + plant seedlings" (34%) for all evaluated methods. The image monitoring methods suggested by this study proved to be efficient, mainly by reducing the time and cost necessary for field monitoring and increasing the accuracy of the generated management recommendations.

**Keywords:** recovery of degraded areas; forest restoration; remote sensing; expert systems

## 1. Introduction

According to the Society for Ecological Restoration [1], ecological restoration is a process that assists the recovery of degraded ecosystems. This process has had a rapid rise worldwide over the last decades [2] driven by the environmental awareness of the population and by the need to environmentally legitimate large companies and farmers. Additionally, the United Nations declared that this decade is dedicated to ecosystem restoration, thus the declaration can be a base for stricter environmental legislations, to accelerate existing global restoration goals (e.g., Bonn Challenge) and to increase the number of projects associated with restoration around the world [3].

In Brazil, ecological restoration projects are increasing, mainly after the approval of the Forest Code, Law 12.651 of 2012. From what was established in the aforementioned law, the Brazilian government approved Decree No. 8972, of January 2017, which establishes the National Policy for Recovery of the Native Vegetation (Proveg). One of its guidelines (Article 6) states that Planaveg (National Plan for Recovery of the Native Vegetation) should contain a plan for monitoring restoration areas that supports decision making aimed at the success of native vegetation recovery.

In this scenario, several projects related to forest restoration have been developed, making it possible to learn through the successes and failures involved in the development of these projects [4]. It is important to monitor restoration areas, since through this monitoring it is possible to assess success or failure [5,6] and to correct the restoration trajectory through the generation of adaptive management recommendations [4,7,8]. The implementation of monitoring and the elaboration of expert systems to help management interventions increases the efficiency of ecological restoration processes and avoids wasting time and resources invested [8,9].

Currently, there are numerous ecological, socioeconomic and management indicators for monitoring restoration described in the Atlantic Forest Restoration Pact [7,8]. The assessment of these indicators using field sampling units is time-consuming and costly, especially for large scale projects, and requires qualified technicians [9,10]. Furthermore, the data processing of large scale restoration projects and the selection of indicators to be measured, paired with their reference values is also complex [4]. Thus, the use of supervised classification of high resolution images, associated with the elaboration of a decision making expert system are promising tools for developing monitoring protocols and for the generation of management recommendations, especially for extensive restoration areas [8–10]. Moreover, these techniques increase the quality and precision of the synoptic analysis and reduce the need for field efforts [11].

According to Shu-Hsien [12], expert systems transfer knowledge from human to computer. This knowledge is constructed mainly with rules that reproduce the expert knowledge, and these rules are usually Boolean or binary sentences [13]. Expert systems are powerful tools to generate solutions to many problems that often cannot be solved by other methods [12]. These systems are widely applied in sectors of our social and technological life, e.g., medical treatment, personal finance planning, engineering failure analysis, waste management, climate forecasting, agricultural management, environmental protection, urban design, but could also be used more frequently for the evaluation of remote sensing applications by generating management recommendations.

The aims of the present study were (1) to build an expert system to generate management recommendations for areas under a restoration process through pre-monitored images from UAV (Unmanned Aerial Vehicle) camera and LiDAR (Light Detection and Ranging) data, (2) to generate management recommendation maps and tables for the studied area, (3) to compare the generated recommendations for each classified image.

## 2. Materials and Methods

### 2.1. Study Site

This study was conducted in conservation areas undergoing restoration on a farm named "Project Maria Mirreis T734" (39.68°S, 17.71°W) owned by Fibria Celulose S.A. (currently Suzano S.A.), located in the southern portion of Bahia state, Brazil (Figure 1). The farm has a total area of 664.42 ha, the majority (432.09 ha) is planted with eucalyptus, 207.17 ha are conservation areas, and the rest includes roads (15.78 ha) and other forms of land use (9.38 ha). The region has a transitional climate, hot and humid tropical climate on the coast and seasonal climate, with dry winter in the interior [14] with an average annual rainfall of ca. 1100 mm [15] and an average temperature of 25 °C, without a defined dry season [16]. The natural vegetation in the region includes the Atlantic Forest (dense ombrophilous forest) [17], which is the target of most restoration efforts.

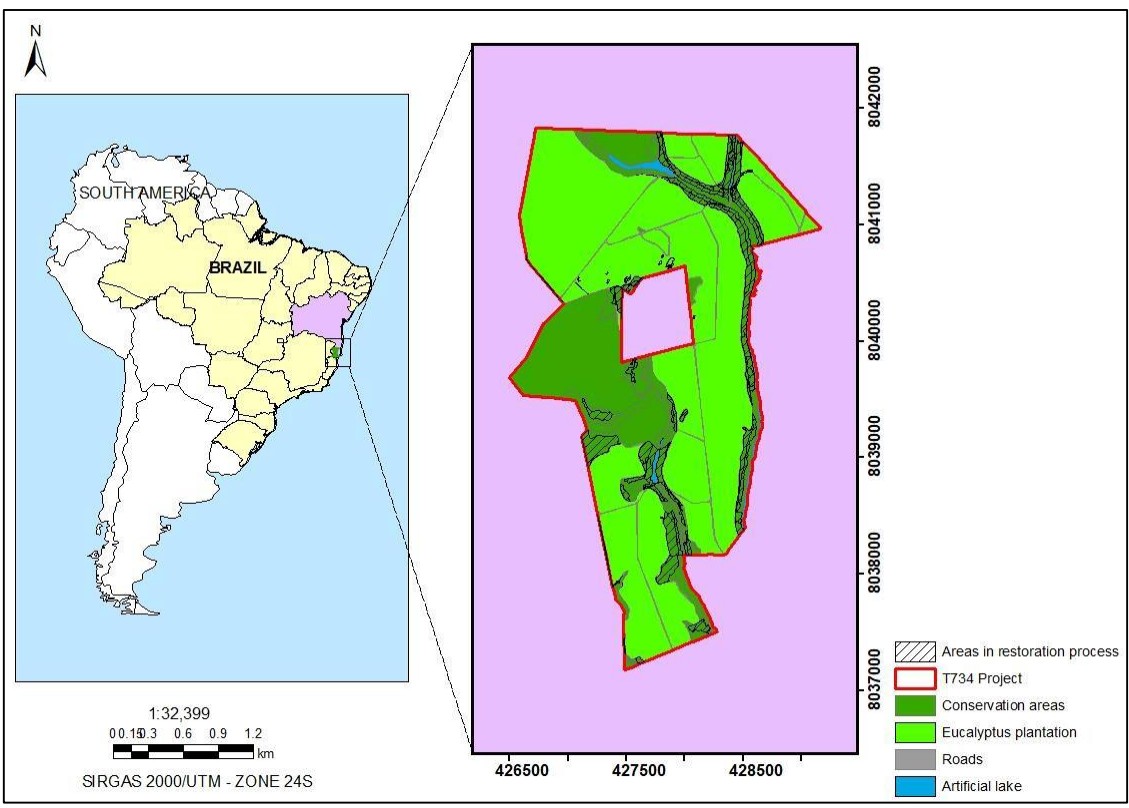

**Figure 1.** Location and land use of the study site called "Project Maria Mirreis T734" in the southern part of the Bahia state, Brazil.

Since 2010 restoration of the Atlantic Forest in the study site was carried out in conservation areas mainly comprised of degraded pastures (54 ha). Among the conservation areas there are natural forests, mainly dense ombrophilous forest, in different successional stages and degraded pastures dominated by the exotic grass *Brachiaria decumbens.* Restoration was conducted either by allowing natural regeneration or planting seedlings of native tree species in a succession-based model which consists of 'filling' and 'diversity' planting lines. According to Rodrigues et al. [4], the filling line consist of fast-growing species which promote fast canopy coverage and improve environmental conditions near the ground surface, hindering the growth of invasive grasses, whereas the diversity line includes slow-growing species and/or low canopy cover, and it is expected that slow-growing species will gradually re14ace those of the filling group when they are in senescence, improving ecological succession in the restored area.

## 2.2. Land Cover Monitoring

The restoration success indicator monitored was land cover. In order to monitor the land cover indicator, the data composition of high resolution aerial images from UAV (Unmanned Aerial Vehicle) and LiDAR (Light Detection and Ranging) cameras was previously classified by the maximum likelihood (ML) and random forest (RF) algorithms [11].

UAV images were acquired in spring with a Canon S110 NIR 12M pixels focal distance/fly height ratio resulting in a spatial resolution of 0.08 m. The camera took images in the spectral region of green, red and near infrared (NIR). Two vegetation indices were determined, Normalized Difference Vegetation Index (NDVI) [18] and Soil Adjusted Vegetation Index (SAVI) [19]. Then, we generated a combined composition with the three spectral bands and the two spectral transformations.

LiDAR data was acquired through a RIEGL LMS—Q680I Airborne Laser system, which works in the near infrared region with a wavelength of 1036 nm. The images were captured in spring. Using LiDAR point clouds, we created a pulse return intensity image, Digital Surface Model (DSM), Digital Terrain Model (DTM) and Normalized Digital Surface Model (nDSM) using ArcGIS 10.2 [20]. The intensity is defined as the ratio of strength of reflected light to the emitted light, and can be useful in classifying land cover [21]. From LiDAR data, a 0.5 m spatial resolution image with four bands (intensity, nDSM, DTM, DSM) was created. Therefore, the image contained the vegetation height and the spectral information associated with the pulse return intensity. For further details about the remote sensing methods see Reis et al. [11].

The land cover indicator was partitioned into three classes: Canopy cover, bare soil and grass cover. These classes were chosen because they can be easily identified through visual image interpretation and they have completely different spectral signatures, which makes the digital image processing more feasible. Additionally, the three classes also have ecological importance in the early stages of the restoration process and are relevant for generating management recommendations. Besides the three mentioned classes, it was necessary to establish another class representing objects in shadows (referred to as "shadow"), because the presence of shadows in UAV images modifies the reflectance of the objects and can reduce the classification accuracy.

Images obtained through the two remote sensing methods (UAV and LiDAR) were classified by Maximum Likelihood (ML) and Random Forest (RF) algorithms and related results, and evaluation of the different methods were published in Reis et al. [11].

## 2.3. Generation of Management Recommendation

We have generated management recommendations based on reference values pre-defined for each indicator class. For canopy cover the reference values were based on the Atlantic Forest Restoration Pact [7] and Viani et al. [8]. Those for the grass cover were based on the company's standard for weed control [22]. For the bare soil cover we selected a reference value as a way to prioritize areas of greater potential to control erosion (Table 1).

**Table 1.** Selected land cover indicators for restoration monitoring with UAV and LiDAR, and categorization defined according to reference values for each class.

| Class/Indicator | Reference Values | Categorization |
| --- | --- | --- |
| Canopy cover | 0–59%<br>60–69%<br>≥70% | Needs intervention<br>Verify intervention by analyzing other indicators<br>Suitable |
| Bare soil | 0–0.09 ha<br>≥0.1 ha | Suitable<br>Needs intervention |
| Grass cover | 0–35%<br>≥35% | Suitable<br>Needs intervention |

Canopy cover above 70% was considered a suitable canopy density, influencing the amount of light entering the forest, water infiltration, seedling development and controlling undesirable grasses [7,8,23], thus making the development of the area more independent of human intervention. When the area has 60 to 70% canopy cover, it should be further monitored, because the area could achieve the 70% by itself without human intervention. Whenever the area has less than 60% canopy cover it would need further restoration interventions. High grass coverage is indicative of invasion, and it must be controlled. If bare soil is found in large spots, it is necessary to restore the vegetation to prevent erosion and nutrient loss [24].

The classes were measured in restoration areas which were under implementation for more than four years (55 ha), according to the company's standard for ecological monitoring [22]. We have determined the area represented by each land cover class (m$^2$ and percentage) through the classified images.

We designed a decision-making flowchart to generate management recommendations, based on cover conditions, which was the base of the expert system. After that we transferred the information from the flowchart to the computer using Python language. We also generated a model builder able to build tables containing the class area (m$^2$ and in %), the area identification code and management recommendations, and maps as outputs.

### 2.4. Comparison of the Generated Recommendations for Each Classified Image

Finally, we compared the results obtained through the recommendations tables generated from thematic images previously classified by the different algorithms, using descriptive statistics. Figure 2 shows a flowchart demonstrating the methodology we used to generate management recommendations.

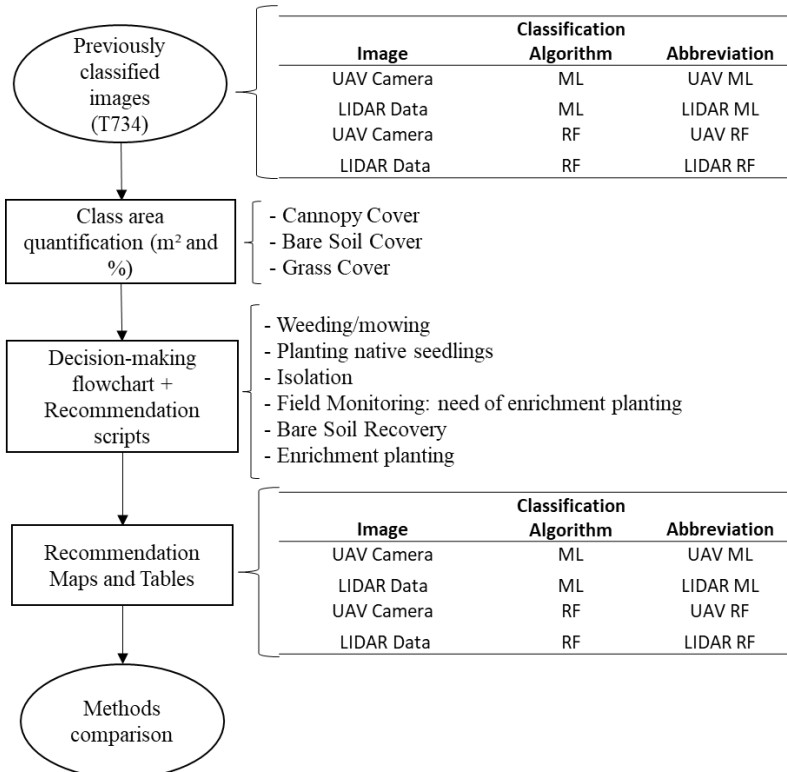

**Figure 2.** Flowchart representing the methodology used to generate recommendations from different classified images. UAV: Unmanned Aerial Vehicle, LiDAR: Light Detection and Ranging, ML: Maximum Likelihood, RF: Random Forest.

## 3. Results

### 3.1. Class Area Quantification

The area covered by the different land cover indicators (canopy, grass and bare soil) in the areas under restoration generated through different image classification methods are represented in Table 2. The different image classification methods had very similar results. Apart from "LiDAR RF", canopy cover had the highest percentage cover in all classified images, followed by grass and bare soil.

**Table 2.** Percentage area of canopy cover, grass cover and bare soil in areas under restoration process found through different image classification methods: UAV camera images classified with ML and RF algorithms, LiDAR images classified with ML and RF algorithms.

| Method | Bare Soil (%) | Grass (%) | Canopy (%) | Shadow (%) |
|--------|---------------|-----------|------------|------------|
| LiDAR RF | 10.6 | 48.6 | 40.8 | - |
| LiDAR ML | 9.6 | 42.0 | 48.3 | - |
| UAV RF | 6.9 | 44.5 | 45.7 | 2.8 |
| UAV ML | 5.6 | 44.8 | 45.4 | 4.1 |

### 3.2. Management Recommendation Elaboration

The decision-making flowchart for the management recommendations that we defined prior to the development of the scripts is presented in Figure 3. In this figure it is possible to observe the indicators selected for monitoring areas under a restoration process (canopy cover, grass cover, bare soil cover), their respective reference values and the recommendations generated for each situation.

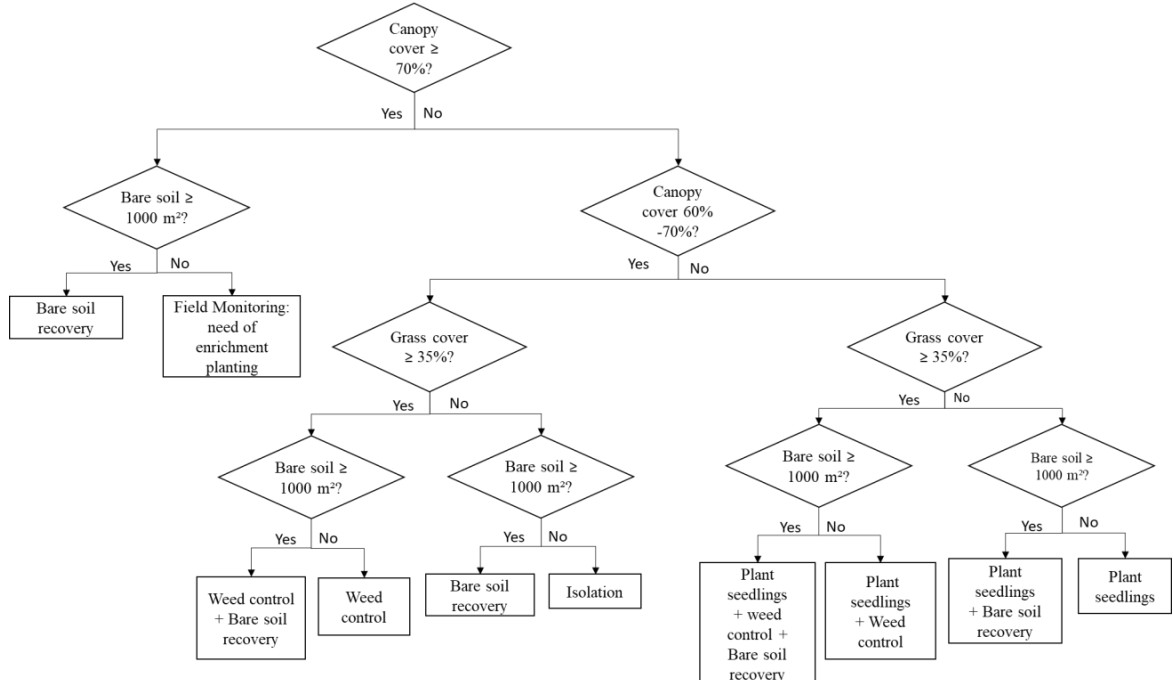

**Figure 3.** Decision-making flowchart to generate management recommendations from the reference values of indicators. Plant seedlings: From the "filling group" list. Weed control: Weeding, mowing.

### 3.3. Management Recommendation Maps and Methods Comparison

The images obtained by a UAV camera and LiDAR data and the management recommendation maps generated from classified images using ML and RF algorithms are presented in Figures 4 and 5, and it represents part of the area under restoration process which was studied in this research. There are

few differences in the recommendations generated with the four distinct methods. These differences can also be seen in Figure 6, which shows the area percentage for management recommendation generated throughout the classified images. Table 3 represents an example of the recommendation tables generated by the model builder. The recommendations that cover the major areas of the T734 project were "weed control + plant seedlings" and "weed control + bare soil recovery + plant seedlings", which represent on average 34% and 29% of the generated recommendations, respectively.

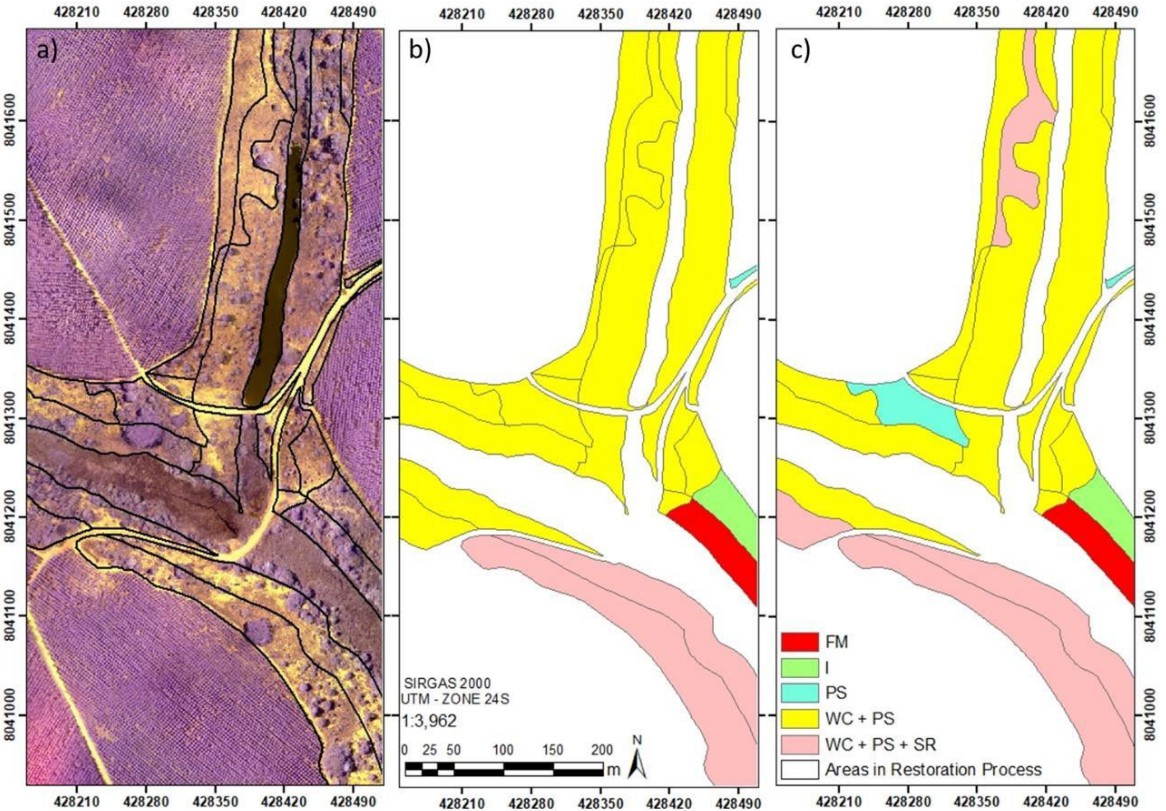

**Figure 4.** (**a**) UAV camera images (red, green, near infrared, NDVI, SAVI), which represent part of the study area; (**b**) management recommendations generated by UAV image classified with maximum likelihood algorithm; (**c**) management recommendations generated by UAV image classified with random forest algorithm. FM: Field monitoring, I: Isolation, WC + PS: Weed control + plant seedling, WC + PS + SR: Weed control + plant seedling + bare soil recovery.

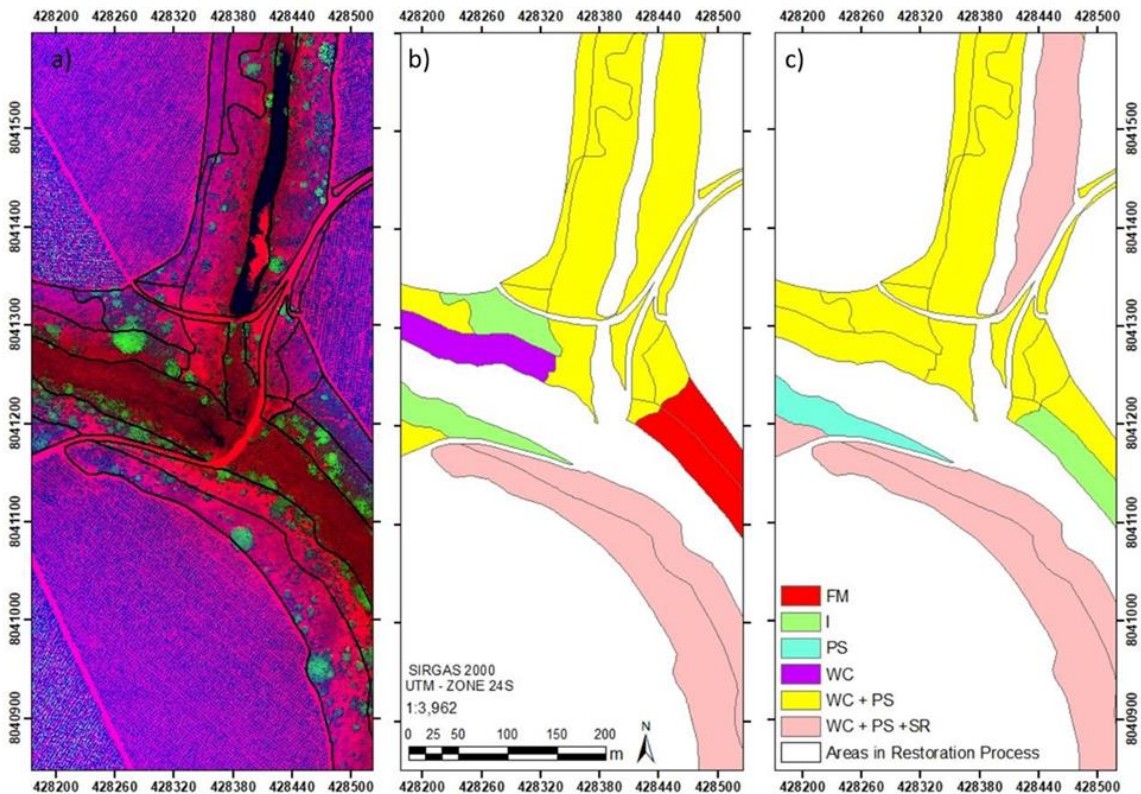

**Figure 5.** (**a**) LiDAR data composition images (intensity, nDSM, DSM, DTM); (**b**) management recommendations generated by LiDAR image classified with maximum likelihood algorithm; (**c**) management recommendations generated by LiDAR image classified with random forest algorithm. FM: Field monitoring, I: Isolation, WC + PS: Weed control + plant seedling, WC + PS + SR: Weed control + plant seedling + bare soil recovery.

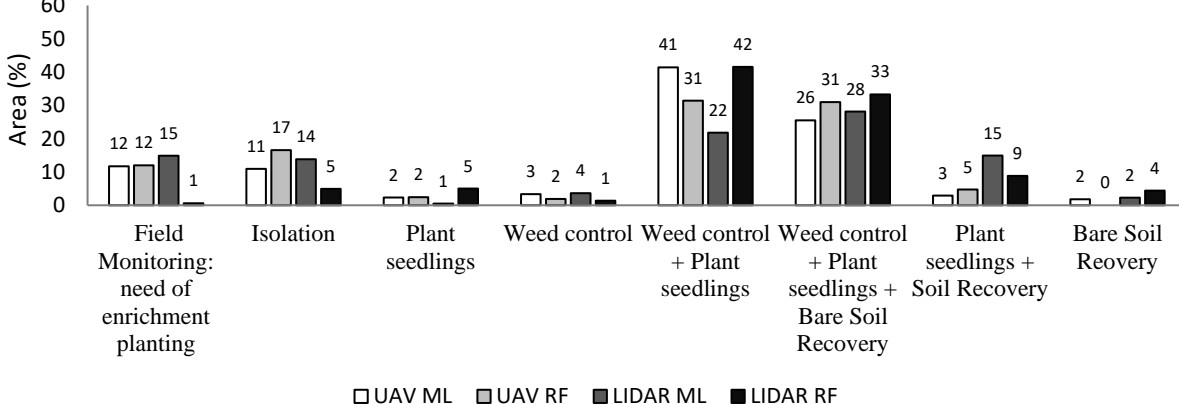

**Figure 6.** Area percentage occupied by each management recommendation according to different monitoring methods, images from LiDAR (Light Detection and Ranging) and UAV (Unmanned Aerial Vehicle), classified by ML (Maximum Likelihood) and RF (Random Forest) algorithms.

**Table 3.** Example of a recommendation table generated by the model builder. The rows represent a certain restoration area inside the farm (T734).

| Location | Situation | Area Total (ha) | Bare Soil (m$^2$) | Canopy Cover (%) | Grass Cover (%) | Management Recommendation |
|---|---|---|---|---|---|---|
| T734/068A | Area in restoration process | 0.79 | 2433 | 28.5 | 37.2 | Weed control + Plant seedling + Bare soil recovery |

## 4. Discussion

Field monitoring and evaluation of several indicators in large-scale restoration projects can be expensive, laborious and time-consuming [9,10,25,26]. However, the use of technologies such as remote sensing and image processing associated with expert systems to monitor land cover indicators has proven to be promising [8], due to temporal and synoptic analysis, precision, reduction of field efforts, time and money savings [10,27]. We elaborated an expert system to generate management recommendation based on images previously classified by different algorithms [11], which proved to be an efficient tool for monitoring restoration.

All four monitoring methods resulted in similar class evaluation with only slight variations. The LiDAR data classified with the RF algorithm was the one that most diverged from the others. These inconsistencies may be related to the image classification process. Differences between LiDAR and UAV may be associated with the presence of shaded areas in UAV camera images (cf. [28,29]). Other differences can be due to erroneously classified pixels in the digital image processing, which need further evaluation. Analyzing the project as a whole, the amount of grass and canopy cover were very similar and high for all evaluated methods, indicating a higher probability to generate more recommendations associated with these classes.

For the areas already well-covered by the canopy, the recommendation was field monitoring to analyze the need of enrichment planting. During field monitoring the density and richness of native tree regeneration should be evaluated, which indicates the increase in biodiversity of the restoration area and the possibility of spontaneous forest succession. Applying remote sensing in the monitoring process reduces the costs associated with monitoring, since it is not necessary to measure these field indicators (e.g., richness and density) in all areas, but only those with a canopy cover greater than 70%. If an area has less than 70% canopy cover it would need further restoration interventions or further monitoring ('isolation'). Restoration activities consist of introducing regional native species from the "filling group" in areas where no seedling development has occurred [4,30], the control of invasive species (weeding or mowing) and/or soil melioration.

The evaluation of invasive grasses in the early stages of the forest restoration process is considered very important. This is because these species compete with seedlings of native species, impeding or hindering their growth [30,31] and spreading rapidly [31]. High coverage by grasses are indicative of invasion, and this must be controlled. Bare soil should also be monitored, because when it is found in large spots it is necessary to recover its superficial layers to prevent seedling death, erosion and nutrient loss [24].

The small variation in the total area of each class estimated by the different methods influenced the generation of management recommendations. About one-third of the recommendations generated for the project included weed control and plantings. This indicates that the methodology of earlier plantings ("filling" and "diversity" planting lines) may not have been adequate for the situation. The low canopy cover is caused by the great mortality of seedlings, mainly species planted in 'diversity' lines, since these have lower tolerance to the competition of grasses in the initial stages of restoration [4,32]. For areas that presented recommendations for bare soil recovery, it would be interesting to use other methodologies to accelerate the recovery process, such as planting supplemented by the addition of green manures or the use of sewage sludge, which accelerate the chemical and physical equilibrium of the soil, making the environment more favorable for the establishment of native species [33].

The image generated from LiDAR data classified with the RF algorithm presented lower canopy cover, and greater cover by grasses and bare soil when compared to others, which generated more recommendations related to weed control, planting and bare soil recovery. According to the study developed by Reis et al. [11], the RF algorithm is more efficient for classifying high resolution images of areas under restoration, as RF (kappa index: 0.94 for LiDAR and UAV) presented significantly higher values of accuracy than the ML (kappa index: 0.88 and 0.90 for LiDAR and UAV, respectively), but the two imaging methods do not differ in accuracy when using the RF algorithm. The choice of one

classification method over the other can lead to higher costs associated with management interventions, however, the accuracy of image classification is more important when deciding on the relevant method.

UAV is considered a low cost alternative for monitoring moderately large areas with prices that range between $300 and a few thousand dollars, whereas LiDAR despite its suitability to monitor larger areas was considered expensive earlier with an acquisition cost of at least $20,000 per flight [9]. These techniques have a rapid development and the prices are fortunately decreasing. According to data provided by Fibria (currently Suzano S.A.) and new researches [e.g., 2], the use of LiDAR is financially feasible for large areas (see also Hummel et al. [34]), and its costs vary from US $1 /ha to US $37 /ha, therefore, increasing the area size can reduce the final cost [35,36]. On the other hand, UAV is feasible for small area imaging [9,27]. In our case, where the area was 55 ha, the price of UAV image acquisition was around $22 /ha (the larger the area, the more feasible the cost). In comparison, field monitoring would cost around $75 /ha, which is more than double the cost of remote monitoring. These values can vary according to the distance between the monitored areas.

## 5. Conclusions

The choice of indicators to evaluate restoration efforts and the definition of their respective reference values remains a challenge for monitoring. This work proposes monitoring based on indicators with rapid and low cost measurements.

The expert system developed in this study, based on remote sensing and the classification of restoration indicators defined according to reference values of each class, is an efficient method that generates results quickly and accurately, helping adaptive management decisions, and allows the reduction of costs associated with field work, especially for large scale projects. Additionally, the choice of the image acquisition method and classifier will influence the final recommendation generated and it can result in higher costs connected to management activities in the short term, however, these expenses will be recovered in the long run, if the selected management methods are more accurate.

**Author Contributions:** B.P.R., S.V.M., T.S.S., E.I.F.F. conceived and designed the research; B.P.R., E.I.F.F., G.E.M. performed the experiments; B.P.R., E.I.F.F., J.M.G., H.G.L., M.H., G.E.M. analyzed the data; T.S.S. contributed with aerial images; B.P.R. wrote the Original draft; All authors wrote-review and edited the manuscript; S.V.M., H.M. supervision; T.S.S., E.I.F.F., J.M.G., H.G.L. were co-supervisors.

**Funding:** The first author and second were supported by the National Council for Scientific and Technological Development (CNPq), Brazilian Government. The last author was supported by the National Research, Development and Innovation Office in Hungary (Project Number FK127996). The field work and data provision was funded by Fibria Celulose S.A. (currently Suzano S.A.).

**Acknowledgments:** The authors of this study thank Fibria Celulose S.A. (currently Suzano S.A.) for providing the data and financial support and the National Council for Scientific and Technological Development (CNPq) for granting the first author's Master's scholarship and research productivity for the second author, and two anonymous reviewers for comments on the manuscript.

**Conflicts of Interest:** The authors declare no conflicts of interest.

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
