# Peer review of "Management Recommendation Generation for Areas Under Forest Restoration Process through Images Obtained by UAV and LiDAR"

_remotesensing, doi:10.3390/rs11131508_

Round 1

Reviewer 1 Report

With this version the authors have addressed all of my majors concerns.

Reviewer 2 Report

The paper has been significantly improved and the reviewers comments were addressed in this version.

This manuscript is a resubmission of an earlier submission. The following is a list of the peer review reports and author responses from that submission.

Round 1

Reviewer 1 Report

The contribution of the manuscript is not clear. The paper needs a literature review.

The writing is incomprehensible. For example in Lines 63: it’s not clear: “It is important to monitor areas that are restoration process, since through this monitoring it is possible to assess success or failure…”

The citation needs revision. For example Atlantic Forest Restoration Pact, 2013; Viani et al., 2017 are cited twice in one sentence (line 63) and many times in one small paragraph.

Line 127 : add a reference.

Line 134: Lidar collect points cloud not imagery.

Have you implemented the decision tree (figure 2)? Implementation needs to be included and thresholds in the decision making  flowchart needs justification.

Reviewer 2 Report

The manuscript researched an important topic - how to cost effectively monitor restoration with increasingly limited budgets which many scientists and restoration practitioners are struggling with. I believe the current state of the article is a good first step in providing a significant scientific contribution to restoration monitoring however I believe that major modifications are warranted before I could recommend this article for publication in Remote Sensing. I have four main concerns related to study design which affect the results, discussion, and conclusions of the article. Firstly, I feel strongly that as presented the article does not truly use adaptive management. In order to fully use adaptive management you need to have a feedback loop where you learn from management decisions and that is the “adaptive” part of this approach. As currently configured the article uses a decision tree approach for what types of restoration should be applied but fails to have, as mentioned above, the feedback loop to test these management approaches and how they actually affect the forest restoration process.  Therefore, if the term adaptive management is to remain in the article I strongly recommend that the authors flush out the additional steps of an adaptive management approach. There is a large body of literature that can provide more details on how to affectively use adaptive management so I point the authors to this body of research.

Secondly, there is much inference and discussion regarding the time and cost effectiveness of the image classification approach to monitoring but there are no actual numbers. For example, how much more cost effective and time effective is this approach versus traditional field based assessments? I believe the most effective study design would have a direct comparison to the field sampling however if such data is not available, as a minimum, other studies should be used as at least an estimate for what these traditional methods take versus the remote sensing methods outlined in the this article (e.g., the remote sensing approach is 50% cost/time effective than traditional field based monitoring). In summary, true numbers as percentage of savings that the method provides would greatly improve the utility for forest restoration practitioners.

Thirdly, there are several instances in the article where the authors state high precision and accuracy of the data however there is no quantitative accuracy assessment presented. I highly recommend that an error matrix be developed and presented showing the quality of the classification versus what exists on the landscape. I feel that this is a requisite for any remote sensing application and without it the readers have no way of knowing how effective the approach is.

Fourthly, the results of the paper are not very compelling because the findings currently show only slight differences between the methods and there is no incorporation of adaptive management feedbacks or accuracy assessment information or true time/cost effectiveness thus if the above three concerns are addressed the results will be more compelling and will be more of a contribution to cost effective forest restoration monitoring.  

Finally, I felt like the article was clearly laid out and easy to follow however especially in the introduction, methods and results there was significant English language edits that were required. I attempted to address as many as I could in the time that I have allotted for this review which I will detail out elsewhere however I recommend that the article is further reviewed by someone proficient as a English language copy editor.

I enjoyed reading the article and having the opportunity to review it. I believe it can be a significant contribution if the above comments are genuinely addressed. 
